# SARS-CoV-2—The Role of Natural Immunity: A Narrative Review

**DOI:** 10.3390/jcm11216272

**Published:** 2022-10-25

**Authors:** Sara Diani, Erika Leonardi, Attilio Cavezzi, Simona Ferrari, Oriana Iacono, Alice Limoli, Zoe Bouslenko, Daniele Natalini, Stefania Conti, Mauro Mantovani, Silvano Tramonte, Alberto Donzelli, Eugenio Serravalle

**Affiliations:** 1School of Musictherapy, Université Européenne Jean Monnet, 35129 Padova, Italy; 2Guzzardi Hospital, 97019 Vittoria, Italy; 3Eurocenter Venalinfa, 63074 San Benedetto del Tronto, Italy; 4Independent Researcher, 44123 Ferrara, Italy; 5Physical Medicine and Rehabilitation Department, Mirandola Hospital, 41037 Mirandola, Italy; 6ARPAV (Regional Agency for the Environment Protection), 31100 Treviso, Italy; 7Cardiology Department, Valdese Hospital, 10100 Torino, Italy; 8Independent Researcher, 60124 Ancona, Italy; 9Independent Researcher, 42023 Reggio Emilia, Italy; 10Istituto di Medicina Biologica, 20129 Milano, Italy; 11Environment and Health Commission, National Bioarchitecture Institute, 20121 Milano, Italy; 12Independent Medical Scientific Commission, 20131 Milano, Italy; 13ASSIS Association, 56123 Pisa, Italy

**Keywords:** COVID-19, SARS-CoV-2, natural immunity, cellular immunity, vaccine-induced immunity, hybrid immunity, cross-reactivity, Omicron

## Abstract

Background: Both natural immunity and vaccine-induced immunity to COVID-19 may be useful to reduce the mortality/morbidity of this disease, but still a lot of controversy exists. Aims: This narrative review analyzes the literature regarding these two immunitary processes and more specifically: (a) the duration of natural immunity; (b) cellular immunity; (c) cross-reactivity; (d) the duration of post-vaccination immune protection; (e) the probability of reinfection and its clinical manifestations in the recovered patients; (f) the comparisons between vaccinated and unvaccinated as to the possible reinfections; (g) the role of hybrid immunity; (h) the effectiveness of natural and vaccine-induced immunity against Omicron variant; (i) the comparative incidence of adverse effects after vaccination in recovered individuals vs. COVID-19-naïve subjects. Material and Methods: through multiple search engines we investigated COVID-19 literature related to the aims of the review, published since April 2020 through July 2022, including also the previous articles pertinent to the investigated topics. Results: nearly 900 studies were collected, and 246 pertinent articles were included. It was highlighted that the vast majority of the individuals after suffering from COVID-19 develop a natural immunity both of cell-mediated and humoral type, which is effective over time and provides protection against both reinfection and serious illness. Vaccine-induced immunity was shown to decay faster than natural immunity. In general, the severity of the symptoms of reinfection is significantly lower than in the primary infection, with a lower degree of hospitalizations (0.06%) and an extremely low mortality. Conclusions: this extensive narrative review regarding a vast number of articles highlighted the valuable protection induced by the natural immunity after COVID-19, which seems comparable or superior to the one induced by anti-SARS-CoV-2 vaccination. Consequently, vaccination of the unvaccinated COVID-19-recovered subjects may not be indicated. Further research is needed in order to: (a) measure the durability of immunity over time; (b) evaluate both the impacts of Omicron BA.5 on vaccinated and healed subjects and the role of hybrid immunity.

## 1. Introduction

COVID-19 is an infectious disease that represents a major challenge infectious disease to human wellbeing; it directly affects health systems, and indirectly involves the economic, politic and social spheres under the form of pandemics [1]. Both natural immunity [2,3] and vaccine-induced immunity [4,5] may be useful to reduce the mortality/morbidity of this disease, but still a lot of controversy exists on the best strategy to manage this complex socio-health issue.

Currently, research on COVID-19 immunity regards the various aspect of the natural as well as the vaccinal immunity (antibody-type, cellular-type), the possibility of relapses after infection and/or vaccination, the immunological memory, the frequency and severity of reinfections, the comparison between vaccinated and unvaccinated populations as regards the type developed immunity, the potential adverse effects of vaccination.

Scientific research on COVID-19 has deeply focused on the specific or adaptive effector immunity, whereas there is a limited knowledge concerning the modulation of this immunity, both of intrinsic (self-limiting) and of extrinsic type (determined by specialized cells of the myeloid and lymphoid kind).

Immunity is considered a well-structured and calibrated process based on mechanisms and functions evolved throughout millions of years [6,7]. Thus, the evaluation of natural immunity should include adjunctive parameters to the classical determination of the antibody titre. In fact, the complexity and multi-modality of the immune reaction to SARS-CoV-2 is being highlighted in several papers in the last two years, though the available evidence on the basic differences between the natural and the vaccine-induced immunity is still limited.

We analyzed all the known aspects about natural immunity against COVID-19, taking into consideration the evolution in the literature throughout these two and a half years, accounting the different viral variants appeared during time.

Our narrative review regards most of the issues in the field of natural and vaccine-induced immunity in COVID-19, starting from the available evidence in the published literature from the beginning of the pandemic until the last months.

## 2. Aims

Due to the several uncertainties which regard the main clinical and cellular/biochemical differences between these two forms of immunity, the present narrative review aimed at eliciting the efficacy of the following three types of immunity within the general population: (a) natural immunity, (b) vaccine-induced immunity and (c) hybrid immunity (vaccinated subjects who are affected by COVID-19). In view of the need for a better understanding of the clinical and cellular/biochemical differences between these three forms of immunity, the present narrative review aimed at analyzing the pertinent literature in order to highlight the development and the consequent efficacy of these types of immunity within the general population.

More in detail, through this review a few specific issues were analyzed: (a) the duration of natural immunity; (b) the type of cellular immunity; (c) the cross-reactivity from other coronaviruses; (d) the duration of post-vaccination immune protection; (e) the probability of reinfection and the related clinical manifestations in the subjects who got COVID-19; (f) the comparisons between vaccinated and unvaccinated subjects in terms of development of immunity and therefore of possible reinfections; (g) the role of hybrid immunity; (h) the effectiveness of the natural and vaccine-induced immunity against Omicron type infection; (i) the typology and incidence of the adverse effects after vaccination in the subjects who previously got COVID-19 compared to the COVID-19-naïve subjects.

## 3. Material and Methods

A literature search was performed to retrieve the published articles regarding natural and acquired (after vaccination or after an infection) immunity with regards to COVID-19. The pertinent articles and documents were collected from a series of scientific search engines: MEDLINE, Google Scholar, PubMed Central, EMBASE, Cochrane Library, ChemRxiv, MedRxiv, BioRxiv, Preprints, ResearchGate, Chemical Abstract Service.

The words COVID-19 and/or SARS-CoV-2 were combined with the following keywords: immunity, immune system, natural immunity, infection, leucocytes, lymphocytes, antibodies, vaccine, vaccination, recurrence, relapse, reinfection, hybrid immunity, spike protein, B-cells, T-cells, cross-reaction, mortality, epidemiology, clinic, Omicron, Omicron 5, Omicron BA.1, Omicron BA.2, Omicron BA.5.

We investigated the available COVID-19-related literature since April 2020 through July 2022, and moreover we took into consideration also the previously published articles where the basic concepts related to the main topics covered in this review were reported (e.g., regarding natural and vaccine-induced immunity).

Nearly 900 in vitro and in vivo studies, mostly on humans, were collected and reviewed; subsequently, we extrapolated the pertinent 246 articles, which constitute the scientific literature on which our narrative review is based.

## 4. Results

### 4.1. Duration and Type of Immunity from Previous SARS-CoV-2 Infection

The studies about natural immunity after COVID-19 infection begun in 2020, and showed a variable duration lasting immunity [8,9,10,11,12]. Already in the first part of 2021 the presence of antibodies for at least 8 months was clear [13,14,15]. Two studies conducted until September 2021 reported the long-term humoral and cellular immunity findings in patients who were affected by COVID-19 and followed up for more than 1 year after the initial SARS-CoV-2 infection, in order to characterize in details the long-term humoral as well as cellular immunity. Both SARS-CoV-2-specific T cells and antibodies could be detected for a period of more than 1 year after infection [16,17]. By the end of 2021, the persistence of neutralizing antibodies one year after SARS-CoV-2 infection in humans was highlighted by other authors as well [18]. Fundamentally, SARS-CoV-2 features 4 structural proteins: spike (S) protein, membrane (M) protein, envelope (E) protein, and the nucleocapsid (N) protein. As to literature data, the most relevant immunogenic role as been attributed to S and N proteins. Protein S especially appears to be the central antigen capable to induce a “protective” host cellular/humoral immune reaction. It specifically stimulates the formation of the neutralizing antibodies (nAbs), which play a central role in the pathogenicity and transmissibility of the virus.

In a large epidemiologic study [19] 39,086 specimens were collected nationwide (USA) and the seropositivity rate was analyzed. This study was performed through the access to a large database of longitudinal data regarding patients recovered from COVD-19. The authors demonstrated the presence of both anti-S and anti-N IgG in the blood samples, and this finding was evident also 300 days post-infection. More specifically, there was an average seropositivity for N-protein in 68% of the subjects after 293 days and a 87% seropositivity of antibodies to S-protein at 300 days. Furthermore, the authors demonstrated that the subjects under the age of 65 had a higher antibody seropositivity.

Another study [20] demonstrated, in a cohort of 214 patients (in asymptomatic, mild to severe forms) recovered from COVID-19, the presence of neutralizing antibodies for a period of more than 480 days. In this study it was also shown that the antibody-dependent immunity can regard also the currently circulating virus variants. In a cross-sectional study of unvaccinated adults [21], antibodies were detected, respectively, in 99% of individuals who reported a positive COVID-19 test, in 55% of subjects who referred a probable COVID-19 contagion without being tested and, finally, in 11% of subjects who referred no specific symptoms or signs of COVID-19 infection. In the same study anti-Receptor Binding Domain (anti-RBD) levels were observed after a positive COVID-19 test for a duration of nearly 20 months.

Additional publications report similar outcomes concerning the typology and duration of natural immunity after contracting COVID-19. Specifically, De Giorgi et al. [22] detected the presence of neutralizing IgG antibodies in a sample of 116 individuals 11 months after the infection, confirming the presence of an immunological memory. In many other publications the presence of SARS-CoV-2-specific humoral and cellular immunity in COVID-19 convalescent subjects was detected [23,24,25,26,27,28]. Of interest, in the work of Wei et al. [2], a randomized sample of 7256 UK citizens previously affected by COVID-19 (with up to 12 months of follow-up) showed the presence of protective antibody levels against SARS-CoV-2 after about 1.5–2 years, as they demonstrated the presence of anti-spike protein IgG antibodies with an average life of about 184 days. Another study focused on the typology of the immune response to SARS-CoV-2 in a selected population of 203 patients recovered from an asymptomatic-to-severe disease [29]. It was shown that 99% of the cases featured the presence of antibodies against the virus, and in 90% of individuals the presence of T CD8 HLA-A2 lymphocytes, specifically directed against the virus.

Other studies, albeit on smaller samples [30,31], have shown the presence of anti-SARS-CoV-2 S-RBD IgG at 14 months in recovered patients. More recently, an interesting meta-analysis included 54 studies from 18 countries for a total of 12,011,447 individuals with 8-month (average) post-infection follow-up [32]. Overall, the authors demonstrated in these subjects the presence of IgG, CD4+ T lymphocytes and B memory cells in 90.4%, and 80.6% of the cases, respectively; moreover, the prevalence of a reinfection was 0.2%.

Other studies showed that the formation and persistence of anti-SARS-CoV-2 B memory cells and of intramedullary plasma cells (which are responsible of humoral immune protection) has been detected in convalescent patients which remain stable for more than 8 months after healing [33]. It has also been documented that there are mutations in the B memory cell compartment, which continue to evolve in the 12 months post-infection [34]. Furthermore, the same mutations were shown to sustain a lasting protection by the memory cells, keeping the germinal centers always active. The presence of persistent antigens has been also demonstrated in other locations, such as the intestine [33]. This specific finding is linked to the constant evolution of antibodies in the germinal centers, which is maintained over time, and which strengthens the immune memory. Recent studies have also documented the presence of IgAs on the surface of the nasopharyngeal mucosa which appear to have the ability to neutralize the infection in the upper airways for several months [34,35] (Figure 1).

A very recent retrospective and large study analyzed the entire Swedish population, demonstrating the presence of natural antibody and cellular immunity capable of protecting from hospitalization after about 20 months. In fact, natural immunity was associated with a 95% lower risk of SARS-CoV-2 reinfection and an 87% lower risk of COVID-19 hospitalization compared to the non-previously infected subjects, for up to 20 months [36]. To prevent one reinfection in the natural immunity cohort during the follow-up, 767 individuals needed to be vaccinated with two doses. In the same study, vaccination was shown to reduce the risk of contracting COVID-19 and hospitalization for up to 9 months, although the differences in absolute numbers, especially as to the hospitalization rate, were small.

Other data suggested that more than 90% of seroconverters make detectable neutralizing antibody responses. Furthermore, it was shown that these titers remain relatively stable for several months after COVID-19 infection [37,38,39]. An older study regarding SARS-CoV infection already showed the presence of SARS coronavirus-specific T cells in three SARS-recovered individuals at 9 and 11 years follow-up. It was also shown that all the detected T memory cell responses targeted the SARS-CoV structural proteins. Furthermore, these responses were found to persist up to 11 years post-infection [24].

In general, the evaluation of the immune response has been predominantly focused on the circulating cells. Recently, a few researchers highlighted an active and crucial role of the cell populations present in some organs and tissues, such as lungs and lymph nodes, in coordinating the persistence of immune memory between the cellular and humoral compartment against SARS-CoV-2. This cell-based immunity was also proven to have a preventive role within the site-specific protection from future infections [40].

Concerning COVID-19 cell-based immunity, still some uncertainties remain. However, the possibility of a very early and effective activation of cellular immunity associated with a complete resolution of the infection, so early as not to elicit any measurable serological response, has also been hypothesized for SARS-CoV-2 [41]. This issue will be highlighted and discussed in the next section.

### 4.2. Cellular Immunity

In addition to antibody immunity, cellular immunity is fundamental in any infectious disease. In fact, whereas circulating antibodies decay over time, cellular immunity is usually maintained active longer in order to produce antibodies when necessary for the same pathogen.

In the assessment of the immune function in COVID-19 at long term, the presence of the neutralizing antibodies was identified as a primary source for protection; conversely, the role of the cellular response, both after vaccination and natural infection, was initially neglected [42]. As per the basic immunology notions, the immune cell response to viral infections notoriously plays a crucial role in limiting clinical progression and in the protection against subsequent infections [43,44,45,46,47,48] which applies to SARS-CoV-2 infection as well. In fact, similar to many other viral infections, COVID-19 was found to be efficiently controlled in most infected individuals through the coordinated activation of the innate and adaptive components of the immune system.

During the SARS-CoV-2 epidemic waves preceding the Omicron variants, an impairment of the Interferon-α (IFN-α) function was found in severe cases, which was documented as mediated by the increase in the production of autoantibodies directed against IFN-α [49]. Conversely, individuals who were mildly symptomatic were able to rapidly develop both a virus-specific antibody and T-cell response, as reported in several scientific studies [50,51,52,53,54,55].

The duration of the follow-up regarding the duration of the immunity after the SARS-CoV-2 infection is getting increasingly longer: the presence of CD4+ and CD8+ T-lymphocytes has been confirmed over time in subjects recovering from SARS-CoV-2 up to 18 months after infection, as reported in a few recent publications [56,57,58,59]; furthermore this T-lymphocyte-based immunity was shown to occur regardless of the severity of the clinical picture related to the infection itself [60,61]. Interestingly, no statistically significant differences between the effectiveness of the immune response to natural infection or to the hybrid stimulation (vaccination + natural infection) was documented after about 20 months [36]. This finding confirms the valid antiviral protection put in place by our immune system over time, after SARS-CoV-2 infection. In these patients the circulating memory of the T CD8+ leukocytes also includes cells with a memory phenotype which is similar to that of stem cells, with sustained polyfunctionality and proliferation capacity. Consequently, these immune cells are likely to play a crucial role in supporting an anamnestic response [62].

A few studies focused on the possible difference between humoral and cellular immunity in COVID-19. It was observed that the antibody titer decreases more rapidly over time than T cell concentration, and the IgG level has not been found associated with the disappearance of SARS-CoV-2 specific B cells [33,63,64,65]. More importantly it was documented that the spike-specific B cells have been detected for longer periods of time even in elderly patients with rapidly declining neutralizing antibody levels [66].

The value of maintaining an immune response over time in COVID-19 was repeatedly highlighted, which is considered even more beneficial as the immunological defenses proved effective also against different viral variant, including Omicron ones, which, as demonstrated, show an important immuno-evasion activity to the currently available vaccines [67,68,69]. With reference to this durable immune condition, a recent study suggested that T cells may target different regions of the Spike protein, including those that are not involved in major mutations. Moreover, these mutations seem to decrease the neutralizing action of the antibodies produced in response to vaccination [70,71].

Among the various T-lymphocytes populations involved in the immune response in this infection, the CD4 lymphocytes seem to play a major role; hence the preservation of T CD4-cell mediated immunity against SARS-CoV-2 is critical for reducing disease severity, as demonstrated by the importance of a rapid T-cell response in preventing severe COVID-19 [51,72,73,74].

Recently, a study measured the efficacy of CD4+ T-lymphocyte immune function against Sars-CoV-2. The related findings indicate that efficient early disease control also predicts favorable long-term adaptive immunity [75].

Furthermore, a durable form of B cell immunity is maintained even if circulating antibody levels decline in time [66].

### 4.3. Cross-Reactivity

In addition to the natural immunity that follows the primary viral infection and protects against possible relapses, the phenomenon of cross-reactivity tends to occur when the immune system identifies proteins in two different agents as similar, thus reacting against both of them. The phenomenon of cross-reactivity of T cell immunity was already known in the past for other acute infections [76]. With regard to the H1N1 flu, cross-reactivity has been demonstrated by those who had already contracted the virus of swine origin [43]. At the same time, neutralizing T CD8+ lymphocytes were found in patients who had had H1N1 infection and who were subsequently protected from symptomatic flu episodes [44].

Another study found T CD4 lymphocytes from previous influenza viruses were capable of mitigating other viral infections [45]. Instead, influenza vaccines did not allow the development of cross-reactivity towards the H1N1 virus [77]. Basically, cross-reactivity seems to be an exclusive phenomenon occurring within natural immunity. This feature was even found in survivors of the Spanish flu, who 90 years later still had circulating B cells capable of producing antibodies. In this case, cross-reactivity was demonstrated against viral agglutinins from 1930’s swine flu [78]. The reactivity of T lymphocytes against SARS-CoV-2, which was present in 20–50% of people with no documented exposure to the virus, was early studied in 2020 [79,80].

Another study detected SARS-CoV-2 reactive T CD4 cells in 40–60% of individuals who were not exposed to the virus, suggesting the recognition of cross-reactive T cells between circulating cold coronaviruses and SARS-CoV-2 [81]. Additionally, cross-reactivity has also been demonstrated following previous beta-coronavirus infections [82] and cellular immunity of T lymphocytes from other coronaviruses has been investigated by several research groups as well [41,71,82,83,84]. Figure 2 shows the possible different mechanisms at the basis of cross-reactivity immunity elicited through T-lymphocytes.

An immunological imprinting by previous seasonal coronavirus infections that can potentially modulate the antibody profile to SARS-CoV-2 infection was similarly demonstrated [85]. Lastly, cross-reactivity of T lymphocytes was also found starting from cytomegalovirus (CMV) and influenza viruses [86,87]. It seems cross-reactivity is a phenomenon which is equally distributed between different genders and ages, although it is more common in children [88], and furthermore this beneficial immunological memory was found of clinical relevance in terms of mitigation of SARS-CoV-2 infection [89].

In fact, this immunological condition had already been hypothesized for COVID-19 patients in 2020, when it was clear that more investigations would be needed [90,91]. From this point of view, an important study [92] has shown that the immune activity stimulated by other coronaviruses (HCoV) is associated with higher immune responses to SARS-CoV-2, indicating a cross stimulation. Above all, HCoV immunity was reported to affect the severity of the disease, since patients with high HCoV reactivity were less likely to require hospitalization. Beyond the cellular immunity of T lymphocytes, also some antibody-based immunity against SARS-CoV-2 deriving from B cells has been demonstrated in subjects previously infected by different coronaviruses [10,93,94].

Lastly, cross-reactive antibodies of both IgG and IgA type have been found also in patients with mild COVID-19. In the study, IgG and IgA to HCoV are significantly higher in asymptomatic than symptomatic seropositive individuals. Thus, has been hypothesized that pre-existing cross-reactive HCoVs antibodies could have a protective effect against SARS-CoV-2 infection and COVID-19 disease [95] In this study, HCoV-derived IgG and IgA have been found significantly higher in asymptomatic than symptomatic seropositive individuals. Thus, it has been hypothesized that pre-existing cross-reactive HCoVs antibodies could have a protective effect against SARS-CoV-2 infection and COVID-19 disease. Figure 3 shows the complex interplay among several immune system elements which are involved in the cellular/humoral response to wild pathogens and reinfection of homologous or variant pathogens (e.g., SARS-CoV-2).

It has been widely determined that bacteria such as Klebsiella pneumoniae, Acinetobacter spp. and Pseudomonas spp., Escherichia coli and Staphylococcus spp., are the most frequently detected additional pathogens in COVID-19 patients [96,97]. This potential secondary cross-infection between SARS-CoV-2 and respiratory tract bacteria may induce a natural cross-reactivity immunity, and this pose the necessity to better treat these patients, also using tailored antibiotics and probiotics when needed [98,99].

### 4.4. The Duration of Post-Vaccination Immune Protection

Immunity against the Delta variant of SARS-CoV-2 decreases in all age groups a few months after receiving the second dose of the vaccine: about 2/3 of severe COVID-19 cases were reported in individuals who had received two doses of the Pfizer vaccine in a study which was performed in Israel during the early vaccination period [100]. Furthermore, the evidence of long-term protection of vaccines in people under the age of 16 against the multiple variants of COVID-19 is even more limited [101].

It has been also reported a lack of vaccination protection in about 8% of non-responder vaccinated people [102]. Currently, it is not known the additional protection induced by the vaccine over the previously infected people. However, it was documented that following vaccination, the efficacy against infection reaches its peak in the first month after the second dose and then it gradually decreases and reaches about 20% in months 5 to 7 after the second dose; at the same time, protection against hospitalization and death persists at a solid level for 6 months after the second dose [5]. The decline in vaccine efficacy appears to be greater in the elderly people, i.e., those aged 65 and over [103].

The antibody titers decay relatively rapidly after the administration of two doses of the vaccine. Such decreases are faster than the reductions in the induced protection from severe disease [96]. The efficacy against symptomatic COVID-19 infection among 842,974 vaccinated individuals was shown to decay and rapidly tend to vanish after about 6–7 months, possibly becoming even negative for longer time intervals [36]. The ongoing generation of new variants, resulting from the selective pressure exerted by the vaccine on the virus, has been reported by a few authors as a possible explanation for this short-term efficacy of vaccination [104,105]. Moreover, it is known that vaccines induce a spike-protein targeting immune response, and in fact most virus mutations affect just this protein, which may additionally explain the short duration of the protective action of vaccines.

More recently it was demonstrated that vaccination in healed subjects may have little if no epidemiological significance [106,107]. In order to investigate this issue more in depth, since 21 June 2021 German authorities have been collecting data about the rate of symptomatic cases of COVID-19 among fully vaccinated patients. This percentage was increasingly higher and was calculated as 58.9% on 27 October 2021. These figures have basically provided some evidence about the growing relevance of vaccinates as a possible source of transmission. Beyond the German data, other authors have highlighted that fully vaccinated people equally spread SARS-CoV-2 infection [108], showing viral loads similar to unvaccinated individuals. Similarly, the relative need for further checks of the spread of the infection in vaccinated and unvaccinated people was proposed by several authors [109,110].

Infections occurring after two vaccinations and having a viral load peak similar to that of the unvaccinated individuals were also reported with the Delta variant [111]. A major attention over the vaccinated population as a possible and relevant source of transmission was suggested, in order to improve measures regarding public health control [112].

### 4.5. Probability of Reinfection in the Recovered Subjects, and Its Clinical Manifestations

Several studies have assessed the possible effectiveness of natural immunity in preventing COVID-19 relapses. Compared to the cases of primary infection, a recovered subject has a much lower probability to be re-infected [113]. A few key elements have been described within the re-infection issue were described: the probability of reinfection, the duration of natural immunity, the severity of the disease in case of relapse (hospitalizations and deaths) and the antibody concentration.

A series of variables may objectively interfere with the results of the studies which examined this re-infection matter, such as: the size of the analyzed sample, the duration of the study, the methodology of analysis and the of data collection. Early in 2020, two UK-based care units experienced a second COVID-19 outbreak, with 29/209 (13.9%) SARS-CoV-2 RT-PCR-positive cases. In those with prior SARS-CoV-2 exposure, 1/88 (1.1%) individuals became PCR-positive compared to 22/73 (30.1%) with confirmed seronegative status. Another study showed that after four months the protection offered by prior infection against reinfection was 96.2% using risk ratios from comparison of proportions, whereas the protection rate was 96.1% using a logistic regression model [114]. Similar experiences have confirmed relapse rates in the previously infected subjects which varied among 0% [115], 0.11% [116,117], approximately 0.3% [118,119] and 1% [120].

Another study conducted in a selected UK population showed adjusted hazard ratios for reinfection with a baseline positive versus negative antibody test of 0.13 and 0.39, respectively. Of note, 11 of the 12 re-infected participants were symptomatic. Furthermore, the same authors showed that the antibody titers for spike and nucleocapsid were comparable in PCR-positive and PCR-negative cases [121]. Another study performed on healthcare workers in Brazil indicated a relatively high rate of reinfection which was strictly correlated with the lowest antibody responses, but in most cases the data did not formally distinguish between reinfection and re-emergence of a chronic infection reservoir. More specifically, this Brazilian study was performed on a small sample (33 patients) and the risk of reinfection was estimated about 7% [122]. Another, much larger, study [123] found a significantly higher reinfection rate, equal to 10% of the analysed cases; however, in the event of reinfection, the viral load was found about 10 times lower than the one of the primary infection. The 3076 participants were investigated for a period of 6 weeks, which represents a very short follow-up, and the healed subjects were in a small number.

A much lower reinfection rate (0.7%) was conversely found by Kojima et al. [124] (0.7%). A relevant issue in these studies concerning re-infection rate is represented by the possible false positive cases among the new infections, in fact the re-emergence of the primary infection can be traced to the remaining virus within the digestive system [125]. The main data and features of the studies cited above are summarized in Table 1.

Natural immunity seems to be less robust against new variants; in this regard, as per one New-York-based epidemiological investigation [126], the peak of relapses was shown to depend upon the spread of the Omicron variant. Some studies have shown that the presence of a high quantity of antibodies developed following the primary infection guarantees greater coverage from the risk of reinfection [123,125,127]. High levels of antibodies also seem to guarantee lower hospitalization rates [128]. Interestingly, even subjects who have contracted the infection in an asymptomatic form can produce high quantities of antibodies [129].

Overall, there is growing evidence [130,137] concerning the lower severity of the symptoms in case of reinfection, in comparison to the primary infection, with a lower degree of hospitalizations and almost no related deaths. For example, among 7173 subjects previously recovered from the infection, 24 cases of reinfection were highlighted, of which 4 required hospitalization (0.06%) and only one subject died [119]. A recent study, published by Crawford and coll. in The Lancet [131], analyzes the cases of relapse in a pediatric population; the lowest reinfection rate has been found in children under the age of 5, i.e., in that age group where vaccination was basically not practiced. This outcome is confirmed by the absence of cases of relapse in minors during the period March 2020-May 2021 [119]. As none of the minors was vaccinated at that time, the post-COVID-19 natural immunity has likely played a fundamental role. Crawford study is of some importance as it featured a long follow-up time (515 days) and demonstrated a very low risk of reinfection, from 0.18% in children under 5 years, up to 0.73% in older than 16 years, in the investigated population.

Of interest, the data reported above are comparable to the ones from the ISS [Istituto Superiore della Sanità—the Italian Higher Institute of Health] report [138,139]. A recent systematic review [140] collected and analyzed 11 cohort studies published during 2020 and 2021 and regarding 615,777 COVID-19 infected subjects, with a follow-up of more than 10 months. From this comprehensive review the following outcomes can be highlighted: (a) reinfection is a relatively rare event (probability of reinfection between 0% and 1.1%), (b) no studies report an increased risk of recurrence over time. On the other hand, it was demonstrated that the protection against contagion provided by vaccinations is both inferior and less lasting than the protective effect of the natural immunity after COVID-19 [141]. More specifically, the authors have also noticed that natural immunity does not vanish at least in the 10 months following the primary infection. A similar conclusion was reached in another trial [136] where it was shown that in the 90 days following primary infection, immunity tends to grow; this finding suggests that natural immunity can last for a very long time.

### 4.6. Comparisons between Vaccinated and Unvaccinated Subjects in the Development of Immunity and Therefore of Possible Reinfections

Several epidemiological studies report about the occurrence of a protection from reinfection and from clinically severe disease in individuals with prior SARS-CoV-2 infection. In particular, two systematic reviews were conducted on the available literature, according to the PRISMA guidelines, in order to determine the effective protection offered by the natural immunity in the general non-vaccinated population [142] and in individuals subjected to complete vaccination course [143]. Specifically, in the review carried out by the group of Kojima et al. [142], the weighted mean reduction in the risk of reinfection was 90.4% with a standard deviation of 7.7%. Protection against SARS-CoV-2 reinfection has been observed for 10 months and was similar to that offered by vaccination [144].

The systematic review by Shenai et al. evaluated observational and randomized controlled trials; all the included studies found at least a statistical equivalence between the protection offered by the complete vaccination and the natural immunity; of note, three of the analyzed studies found the superiority of natural immunity. Nine clinical trials were included in their review and the data concerning COVID-cured, COVID-naïve, vaccinated and unvaccinated patients were retrieved. Three of the trials included in the review were sponsored by the vaccine industries [145,146,147] and reported a relatively small group of healed in the subgroup analysis (3–0.15% of the overall cohort). Among the four retrospective observational cohort studies, one non-sponsored study examined the cumulative incidence of SARS-CoV-2 infection in 52,238 employees of a US healthcare system. This incidence was nearly zero among healed unvaccinated subjects, previously healed vaccinated subjects and also among vaccinated COVID-19-naïve subjects. Furthermore, no statistically significant benefit was found for vaccination in individuals recovered from COVID-19.

One of the most notable prospective observational cohort studies comprised in the review [148], included 6.3 million adults and used a dynamic model with adjustment for age, gender, previous PCR test results, and common risk. This study found excellent vaccine efficacy in the COVID-19-naïve group, which was greater than 92%. Additionally, in this study, protection in the unvaccinated cohort was slightly higher with 94.8%, 94.1% and 96.4% protection against infections, hospitalization and serious illness, respectively. The main limitation of this study is the short observation period (3 months). Another recent study [149] documented that SARS-CoV-2-naïve vaccinees had a 13.06-fold increased risk for breakthrough infection with the Delta variant compared to unvaccinated-previously-infected individuals, when the first event (infection or vaccination) occurred during January and February of 2021. The increased risk was significant for symptomatic disease as well. When investigating the data related to the infection occurring at any time between March 2020 to February 2021, evidence of the waning naturally acquired immunity was demonstrated, though SARS-CoV-2 naïve vaccinees still had a 5.96-fold increased risk for breakthrough infection and a 7.13-fold increased risk for symptomatic disease.

Overall, some of the studies reported in the review of Shenai and coll. have several limitations that may reduce their scientific value. One of the main biases found in these publications is the lack of a systematic PCR test screening of the asymptomatic subjects, which can lead to a possible underestimation of reinfections. Only one study took into consideration serological positivity as a marker of previous infection, whereas in most trials no screening before vaccination was performed. Other studies have a relatively small sample size, or lack of adjustments for baseline demographics [150]; moreover, in one case [150] only the less frequently used ChAdOx1 Nov-19 vaccine was used. Lastly, some trials were conducted during the Delta strain emergency, which led to a reduced average follow-up. However, the authors of the review conclude that previous SARS-CoV-2 infection provided greater protection than the one afforded by the single or double dose vaccine.

Similar conclusions were reached by many more studies [133,151,152,153,154]. Specifically, a recent study has shown a relative 96.7% reduction in the incidence of reinfection by SARS-CoV-2 in the group of recovered unvaccinated patients [135].

A comparative study [155] analyzed the incidence rate of reinfections and hospitalizations in California and New York during the period between May and November 2021. From this analysis it appears that what affects the incidence rate granting immunity is mainly the timing of the last event, i.e., the time elapsed since the infection and/or vaccination. In fact, the hypothesis of a faster decline in protection against SARS-CoV-2 infections in COVID-19-naïve vaccinated than in unvaccinated recovered individuals has been verified by multiple studies [156,157]. Interestingly, literature data [152] show that in vaccinated subjects the initially highest antibody titers decrease by up to 40% each subsequent month, while in convalescents the reduction is about 5% per month. Moreover, it was clearly shown that the BNT162b2 mRNA vaccination elicits a strong systemic immune response by drastically increasing the development of neutralizing antibodies in the serum, but not in the saliva, thus failing to limit the acquisition of the virus upon its entry [158].

The persistence and the neutralizing capacity of the specific antibodies in healed patients with lasting protection was recorded in several longitudinal studies (12 months in the study by Hwang et al. [159]; 13, 14, 18 months in the studies by Gallais et al. [154], Eyran et al. [157] and Dehgani-Morabaki et al. [160]).

It has been furthermore hypothesized that this natural immunity protection can be quite effective also against the latest variants [161,162], as also confirmed through laboratory in vitro tests [163] and especially documented through the data provided by a systematic review [164].

The analysis of the different humoral and cellular responses in these two groups of subjects was also taken into consideration: for example, in the study by the group of Tarke et al. [72], T CD4+ and CD8+ cells specific for SARS-CoV-2 are compared to lineages B.1.1.7, B.1.351, P.1 and CAL.20C in convalescent COVID-19 subjects and in subjects vaccinated with mRNA-1273 or BNT162b2. The authors proved that overall reactivity against SARS-CoV-2 variants is similar in magnitude and frequency of response, with decreases in the 10–22% range observed in some test combinations. Unfortunately, this study does not include the last two Omicron variants (B1 and B2), however one recent study [93] found that in hospitalized patients with Omicron infection there were T-cell responses to spike protein, nucleocapsid and membrane proteins which were comparable to those found in patients admitted to hospital in previous waves dominated by Beta or Delta variants. Therefore, despite Omicron’s extensive mutations and reduced susceptibility to neutralizing antibodies, most responses from the T lymphocytes, induced by vaccination or infection, recognize the variant through a cross reactivity.

In COVID-19-naïve individuals, the second dose of vaccine was shown to increase the quantity and altered the phenotypic properties of SARS-CoV-2 specific T cells. However, in recovered vaccinated patients, T cells exhibit different phenotypic characteristics that suggest a persistent and long-lasting nasopharyngeal localization able to respond robustly to emerging viral variants [165]. Comparing the efficacy of natural and artificial immunity, a recent study found evidence of an increased risk of infection by the Beta (B.1.351), Gamma (P.1), or Delta (B.1.617.2) variants compared to the Alpha (B.1.1.7) variant after vaccination, without clear differences between vaccines. In contrast to vaccine-induced immunity, in the same study there was no increased risk for re-infection with Beta, Gamma or Delta variants relative to Alpha variant in individuals with infection-induced immunity [166].

In a recent trial conducted in Island, the authors estimated the proportion of persons who became re-infected with SARS-CoV-2 during the Omicron wave. The probability of reinfection increased with time from the initial infection (odds ratio of 18 months vs. 3 months, 1.56; 95% CI, 1.18–2.08) and was higher among persons who had received 2 or more doses compared with 1 dose or less of vaccine (odds ratio, 1.42; 95% CI, 1.13–1.78) [167].

Analyzing different vaccinated groups, several studies reported that after a single vaccine injection, the median titer of specific antibodies in individuals previously affected by symptomatic/asymptomatic COVID-19 was found far above the median titer found in COVID-19-naïve subjects undergoing a full vaccination program [168,169,170,171,172,173]. Only one study [174] documented that the levels were similar in the two groups as to above, but this statistical finding was affected both by the numerical difference of the two compared subgroups in favor of the uninfected vaccinated (35 vs. 228) and by the short observation period (3 days after the 1st dose, 7–21 days after the 1st and 7–21 days after the 2nd dose in COVID-19-naïve subjects).

### 4.7. The role of Hybrid Immunity

There are studies that indicate that vaccination in recovered patients increases the antibody titer [164,171,175,176] or improve the outcome of the disease [150,170] but in some cases these studies have been carried out only in vitro, and therefore do not consider the clinical aspects (see for instance [177,178,179]). For example, there are some studies cited by the European Medical Association (EMA) regarding the administration of two vaccine doses to the recovered subjects [148,179,180,181]. Actually, these studies contain various biases, such as the failure or insufficient performance of verification tests for previous infection, and, consequently, the detection of subsequent reinfection. As for reinfections, no clinical data emerges on the rates of asymptomatic and symptomatic cases, which are fundamental for identifying the real clinical need to vaccinate a recovered individual. Finally, the groups are not closed: patients could be transferred from one to another depending on the vaccination or infection status. Therefore, the accuracy of the follow-up time estimates may have been compromised.

In any case, these studies indirectly highlight a few pertinent findings: (a) the protection provided by a previous infection is superior, in terms of duration and efficacy, to the artificial one acquired through vaccination, (b) the use of one or two doses is irrelevant in terms of final protective efficacy, (c) compared to the protection offered by vaccination, which decreases in the short term, the one acquired by a previous infection remains stable for up to 15 months. Oppositely, a recent prospective Italian cohort study [182] proved that the probability of infections after vaccination is significantly lower than reinfections after natural infection. It should be noted that in the same study reinfections were identified as two positive PCR samples, interspersed with a negative PCR, in the same subject after more than 60 days. According to both the CDC and the ISS, by definition, reinfection must take place at least 90 days after the first diagnosis. Alternatively, there must be sequencing that demonstrates the presence of a viral strain different from the previous one. The need to distance the diagnosis of reinfection, due to the possible viral persistence for more than 90 days, has been also highlighted by several authors [132,136,183].

A retrospective cohort study recently published in The New England Journal of Medicine (NEJM) [184] was meant to evaluate the reinfection rates in recovered patients, comparing them with the group of subjects who underwent COVID-19 vaccination. In this study the recovered population was divided into two groups (unvaccinated and vaccinated). This subdivision was dynamic, that is, the participants who were vaccinated remained in the first group (unvaccinated) for the first 7 days after administration, citing the time necessary for the vaccine to prove effective as a motivation. In fact, the same company producing one of the vaccines observed that within the first 7 days of vaccination there is a 43% increase in infections (FDA Briefing Document, [145,146,147]). Therefore, the allocation of the vaccinated subjects for the first 7 days in the unvaccinated group after vaccination in the study cited above renders the authors’ conclusions on the comparison of reinfections in the two population groups debatable.

One more possible bias of this comparative study is represented by the increase in the infection rate in the first seven days following inoculation which could be due to a transient decrease in lymphocytes observed in all ages and in all dosage groups after the first dose. Another limitation of the NEJM study above, as correctly pointed out by the authors, is represented by the significantly lower number of PCR tests performed in the vaccinated group compared to the unvaccinated cohort: in fact, the vaccinated subjects were not systematically tested, but only in the presence of relevant symptoms. In this way, asymptomatic infected people were identified only in the unvaccinated group. Similarly, the authors properly report another limitation of the study, represented by the relevant lack of data on the severity of infections and on hospitalization and death.

Another randomized study demonstrated the reduced risk of reinfection in patients who were previously infected and then vaccinated compared to the unvaccinated. Anyway, there were only 10 hospitalizations overall, so no relevant statistical conclusions can be drawn. There was no COVID-19-related death during the study [185]. Abu-Raddad et al. [186] through a large cohort study of 1.531.736 mRNA-vaccinated individuals in Qatar, found that among BNT162b2-vaccinated persons, 159 reinfections occurred in those with and 2509 reinfections in those without antecedent infection 14 days or more after the second dose. Similarly, among mRNA-1273–vaccinated persons, 43 reinfections occurred in those with and 368 infections in those without antecedent infection. They concluded that prior SARS-CoV-2 infection was associated with a statistically significantly lower risk for breakthrough infection among individuals receiving the BNT162b2 or mRNA-1273 vaccines. A recent comparative trial [134] proved that the risk of reinfections, hospitalizations and deaths is reduced in SARS-CoV-2 reinfections versus primary infections. Furthermore, the authors compared naturally immunized subjects with vaccinated subjects and they concluded that natural immunity may offer equal or greater protection against SARS-CoV-2 infections compared to individuals receiving two doses of an mRNA vaccine, but the published data are not fully consistent. Lastly, the role of hybrid immunity remains unclear, as to their findings.

Most studies agree that there is no significant increase in cellular immunity [187], circulating antibodies, neutralizing titers, or antigen-specific memory B cells in recovered subjects after the second vaccine dose [159,188,189]. When present, this increase is characterized by the rapid decay of the antibody titer [160], concurrent with a greater occurrence of the post-vaccine adverse events. Additionally, in one of the previously reported systematic reviews [143] it was observed that vaccination in subjects recovered from COVID-19 provides modest protection from reinfection (RR = 1.82 [95% CI 1.21–2.73], *p* = 0.004) with an extremely marginal difference in absolute risk (RA = 0.004 person-years [95% CI 0.001–0.007], *p* = 0.02); at the same time, adverse events after vaccine injection were more frequent after the second dose (mean: 0.95 vs. 1, 91) in healed subjects compared to the COVID-19—naïve ones (mean: 1.63 vs. 2.35).

### 4.8. Effectiveness of Natural and Artificial Immunity against Omicron

The sequence of the SARS-CoV-2 Omicron variant (B.1.1.529), which has three major “subvariants” BA.1, BA.2, and BA [190] was first announced on 24 November 2021. This variant has over 30 mutations concerning the Spike protein [191], of which 15 mutations are located in the Binding Domain of the Receptor (RBD), that is, in one of the main targets of neutralizing antibodies [192]. In total, the Omicron variant genome contains 18,261 mutations, from which more than 97% are present in the coding region, and the remaining 558 are detected in the extragenic region [193,194]. Preliminary indications showed that the Omicron variant is highly contagious but less dangerous than the previous ones [195,196,197,198]. There is evidence of a reduced risk of hospitalization for Omicron compared to Delta variant infections [196,197,199]. Interestingly, the (low) risk of hospitalization in children under the age of 10 does not differ significantly between the Delta and Omicron variant. Since the Omicron variant became dominant [200], there have been more infections among children, but of lesser severity.

One study [201] involved almost 652,000 children under the age of five in the USA, and it showed significant reductions compared to Delta in terms of access to the emergency room, hospitalizations (−34%), access to intensive care (−65%) and use of assisted breathing (−85%), which were however rare events. This study comprehensively showed a series of reassuring data which are most useful for estimating disease severity among children.

Omicron variant causes milder disease also in adults and, for example, people aged 60 to 69 have a reduced risk of hospitalization by approximately 75% with Omicron compared to Delta [197]. Similar results were achieved also in a very recent Italian study [202]. In England, out of over 1.5 million cases (over a million with Omicron and 450,000 with Delta), Omicron variant in the unvaccinated was 5 times less lethal than Delta one in all age groups, and about 10 times less lethal in middle-aged subjects.

A number of studies have also clarified that, compared to the previous variants, Omicron has markedly decreased the protective efficacy of both a previous infection and vaccinations [68,192,203,204,205]. However, it was shown that the individuals who have overcome the natural infection are protected from an Omicron infection slightly more than those who have had two doses of the vaccine. The difference, 61.9% versus 55.9%, is not statistically significant, but vaccination protection is known to decline much more rapidly over months than that following a natural infection [206,207], in addition to the lack of the mucosal protection which is typical of vaccines, and which is oppositely conferred by natural infection [158].

Compared with other variants, Omicron has more difficulty in entering lung tissues and is more easily found in the upper airways: this could explain its high transmissibility [197]. Part of this reduced severity is likely to be attributed to the protection of the previous immunity: those patients who previously had a Delta variant infection, when infected with Omicron, have a significantly lower chance of severe disease (62.5% vs. 23.4%) [208]. Hence, it was also speculated that a pre-existing innate cellular immunity, with or without detectable neutralizing antibodies, is likely to continue to protect against severe disease [192]. In particular, it was proven that the protection provided by previous COVID-19 infections against hospitalization or death appears solid, regardless of the variant considered [207,208]. Interestingly, a clinical study documented that a previous ascertained SARS-CoV-2 infection offers some protection against hospitalization and especially a high protection against death in unvaccinated individuals [197].

Furthermore, a previous COVID-19 infection protects against symptomatic reinfection with Alpha, Beta or Delta variants by approximately 90%. Conversely, this protection against reinfection is lower in case of Omicron variant, but it is still around 60% [207,208]. With regard to studies involving vaccinated patients, the results on the protection against contagion from Omicron seem to be currently contradictory. Some studies have analyzed T lymphocytes taken from people who received a COVID-19 vaccine or were infected with a previous variant and found that these T lymphocytes can respond to Omicron. Indeed, while antibody immunity may be short-living, the more resistant T lymphocytes are able to perform a variety of immune functions, including their ability to act as “killer” cells that destroy virus-infected cells and limit the spread of infection [191]. Overall, these specific T CD4+ and CD8+ cells, induced by a previous infection or vaccination [209] provide a broad immune coverage against the Omicron variant as well [210]. Unsurprisingly, a substantial degree of natural cross-reactive immunity between the different variants was also described in both two-dose vaccinated patients and in the infected patients [180] which is possibly due to the robust T Cellular CD4+ and CD8+ response generated by both vaccination and previous infection, rather than to the single antibody response [93].

However, it is objectively difficult to distinguish the protection provided by the pre-existing immunity from the intrinsic less dangerous properties of the Omicron variant. In fact, in South Africa more than 70% of the population of the regions heavily affected by Omicron have had a previous COVID-19 infection. This previous exposure to SARS-CoV-2, as well as COVID-19 vaccinations, enhance the likelihood that the immune system presents T cells which recognize fragments of virus proteins and, together with the induced antibody increase, more easily destroy the infected cells [211]. In a retrospective cohort analysis of the entire population of an Italian region, a few authors followed 1,293,941 subjects from the beginning of the pandemic to the current scenario of Omicron predominance (up to mid-February 2022). After an average of 277 days, they recorded 729 reinfections among 119,266 previously infected subjects (overall rate: 6.1%), eight COVID-19-related hospitalizations (7/100,000), and two deaths.

Importantly, the incidence of reinfection did not vary substantially over time: after 18–22 months from the primary infection, the reinfection rate was still 6.7‰, suggesting that protection conferred by natural immunity may last beyond 12 months. In this cohort the risk of reinfection was significantly higher among females, unvaccinated subjects, and during the Omicron wave. In fact, a markedly higher rate of reinfections was recorded during the first 54 days of the Omicron wave (*n* = 613; 11.4 per day) than during the 317 days of the pre-Omicron period (*n* = 116; 0.4 per day) [212].

Other publications have documented that the Omicron variant, having many more spike protein mutations than the previous strains, can more easily escape the neutralizing possibilities of both the vaccine [190] and the natural immunity resulting from infections with previous variants [204]. In fact, the vaccine efficacy against Omicron has been shown to be significantly lower than that against Delta infection and it rapidly decreases in a few months [183,204]. It is also unclear whether boosting with Omicron-specific vaccines would improve immunity and protection [213].

In addition, in the face of the documented protection given by natural immunity, it is observed through literature data that the vaccine is fundamentally unable to avoid contagion from SARS-CoV-2. In fact, in January 2022, in a WHO Interim Statement [214], the Technical Advisory Group on the Composition of anti-COVID-19 Vaccines, while reporting about the most recent data concerning effectiveness of the vaccines against hospitalization, severe illness and death, declared the need for alternative, different vaccines; actually, the WHO committee indicated the necessary improvement and update of the vaccines, in order to have a high impact on the prevention of infection and transmission, also to stimulate a broad, strong and long-lasting immune response, finally to reduce the need for booster doses.

Literature data show that clusters of Omicron variant infection are described in individuals who had completed the primary vaccination course and carried out the booster dose for at least one month with mRNA vaccines [215]. All of these investigated patients had a symptomatic course of COVID-19 with mild to moderate manifestations. Basically, this further evidence documents that three doses of the mRNA vaccine do not prevent infection and symptomatic disease from the Omicron variant [215]. Furthermore, among infected individuals, Omicron viral load was similar between adults who received 3 or 2 doses of vaccine, which could suggest that the booster dose does not positively affect Omicron viral load [216].

Other recent publications highlighted a series of data which confirm the variability of immune protection against Omicron variants, little if no dependent on the vaccine administration. For example, a longitudinal study [217] did not find large differences in the median duration of viral shedding among participants who were unvaccinated, those who were vaccinated but not boosted, and those who were vaccinated and boosted.

A number of early animal studies were early performed to test vaccination performance on Omicron variants. Overall, these studies suggest that also Omicron-specific boosters offer no advantage over a third dose of current vaccines [213,218,219]. Recently, Windsor et al. [220] identified three antibodies that neutralized all VOCs tested (including Omicron BA.1) and used cryo-EM of these antibodies bound with SARS-CoV-2 spike to suggest ways in which somatic mutation might restore VOC recognition by other antibodies.

After Omicron BA.1 and BA.2, the variants BA.4 and BA.5 emerged more recently, but the literature about their clinical impact is scarce at the moment of the redaction of this manuscript. However, it was documented that the effectiveness of a previous pre-Omicron infection against symptomatic BA.4/BA.5 reinfection, irrespective of symptoms, was 28.3%. The protective efficacy of a previous Omicron infection against symptomatic BA.4/BA.5 reinfection was 76.1%, and against any BA.4/BA.5 reinfection was 79.7%. This means that the protection against BA.4/BA.5 reinfection was modest when the previous infection involved a pre-Omicron variant, but strong when the previous infection involved the Omicron BA.1 or BA.2 subvariants [206,207].

It seems that also BA.5 subvariant exhibits an increased transmissibility and immune escape from neutralizing antibodies generated through previous infection/s or vaccination/s, and have caused numerous re-infections and breakthrough infections [193]. Furthermore, it was shown that BA.5 is resistant against the majority of monoclonal antibodies [205] but more robust data are necessary to assess the real clinical impact of the latest variants on the vaccinated and healed non-vaccinated population.

### 4.9. Incidence of Adverse Effects after Vaccination in the Recovered Compared to COVID-19-Naïve Subjects

Safety issue represents a basic element in any drug administration. In the case of anti-COVID-19 vaccines it was documented that some differences may exist in the risk-to-benefit ratio when vaccinating individuals who were not previously infected by COVID-19, or those who were previously infected. Past literature has clearly highlighted that some antigens, such as the one of chickenpox, have the ability to generate, through various mechanisms (e.g., cross-reactivity, induction of autoantibodies, self-induced tissue lesions due to the activation of Interferon Gamma) a condition of autoimmunity, due to their ability to present the antigen and to overstimulate cells such as the host’s T CD4+ and/or CD8+ cells, putting the integrity of the immune system at risk. This systemic autoimmunity occurs when the host’s immune system is overstimulated by external factors, such as repeated exposure to the antigen, at levels that exceed the system’s self-organized criticality [221].

The work of Levi et al. [222] demonstrated that the antibody response of patients who had had COVID-19 was relevant and protective, thus focusing on the possible hyper-stimulation reaction triggered by additional vaccines. Alternatively, vaccination of recovered subjects could cause the formation of low affinity antibodies which would result in a so-called ADE—Antibody-Dependent Enhancement—phenomenon, above all when again exposed again to SARS-CoV-2. In fact, it was reported the association between more clinically significant symptoms after the first dose of vaccine in subjects previously exposed to SARS-CoV-2 infection [223]. In other well-conducted studies, it was shown that anti-COVID vaccination in patients with a previous infection can exacerbate the systemic response to the vaccine [188,224].

Overall, these authors documented how vaccinated individuals with pre-existing immunity derived from previous COVID-19 infection had a higher frequency and severity of systemic reactions than individuals without immunity from COVID-19 infection.

Krammer and coll [188] specifically demonstrated that antibody titers of vaccinates with pre-existing immunity were 10 to 45 times higher than those of vaccinates without pre-existing immunity at the same time points, after the first vaccine dose (e.g., 25 times as high at 13 to 16 days); similarly they found that the antibody titer exceeded the median antibody titers measured in participants with no pre-existing immunity after the second vaccine dose by a factor more than 6. Although the antibody titers of vaccinates without pre-existing immunity increased by a factor of 3 after the second vaccine dose, no increase in titers was observed in COVID-19 survivors who received the second dose of the vaccine. The same study showed that local side effects occurred with similar frequency among participants with and without pre-existing immunity, whereas systemic symptoms were more common among participants with pre-existing immunity.

Other authors [225], showed an approximately 50% increase in the risk of developing systemic reactions in the recovered and subsequently vaccinated subjects, compared to COVID-19-naïve vaccinated individuals. The risk of local adverse events was also greater (from 20 to 40%) in the recovered vaccinated subjects in comparison to COVID-19-naïve ones. In an experimental study which focused on the possible blood variations related to vaccination [226] it was established through a mathematical model that subjects with symptomatic COVID-19 after vaccination have a higher expected blood viscosity than those who have had asymptomatic COVID-19 or have not had it at all.

An increasing number of publications are recently showing clear-cut evidence about the higher incidence of adverse effects when vaccinating subjects who recovered from COVID-19. For example, a few authors on one side the authors documented a higher frequency of systemic reactions to the vaccine in subjects previously infected with COVID-19 than in those with no documented history of infection; on the other side, an increase in blood pressure compared to subjects without previous exposure to SARS-CoV-2 was found as well. This hypertensive state was encountered both in subjects with already overt hypertension and in subjects who had never reported arterial hypertension [227].

A dose-dependent incidence of local and systemic adverse events after vaccination was recorded in another study which stratified the outcomes as per the population age. More specifically, the incidence was higher in participants under the age of 55 and this finding was attributed to the greater reactogenicity that occurs in younger people [228]. In addition, to other published data regarding a higher incidence of vaccine-induced side effects in patients who overcame COVID-19 [229], a few authors highlighted that these patients previously affected by COVID-19 and then vaccinated had an increased incidence of side effects already since the administration of the first dose of the vaccine, as well as a greater severity of these adverse events compared to COVID-19-naive patients. Furthermore, also after the second dose of vaccine an increase in the side effects and severity was documented as well [169].

Coherently with the previous elicited data, Tré-Hardy et al. [230] documented that adverse events correlated to the first dose of mRNA vaccines are more serious in subjects previously affected by COVID-19 than in seronegative ones.

Finally, it was noted that patients previously infected with COVID-19 and afterwards undergoing vaccination featured a poor immune response at the second dose of the vaccine [188].

## 5. Discussion

Literature data highlight the presence of a relevant immune response in most subjects following exposure to SARS-CoV-2, both among vaccinated and unvaccinated individuals. The presence of both a humoral and cellular response has been highlighted, although not of the same entity; this immunological protection occurs regardless of the symptoms manifested during the possible antecedent infection, regardless of gender and of age. The vast majority of the individuals affected by COVID-19 develop the typical natural immunity, which is both cell-mediated and humoral; the elicited immune response proved as effective over time, capable of providing protection against both reinfection and against its severe symptomatology.

Natural immunity was shown to persist for a long period of time, e.g., a minimum of 12 months; moreover, protective antibodies and memory B cells have been found in many follow-ups from 12 months to 20 months after healing from COVID-19. Most authors agree on the probable prolongation of this immunologically protective state over time. This occurs both because of the number of antibodies and thanks to the presence of memory B cells in multiple loci (for example in the bone marrow and intestine), which are constantly evolving, in favor of a long-lasting immunological memory.

The presence of highly immunogenic anti-spike and anti-nucleocapside antibodies has been demonstrated in the healed unvaccinated subjects, which is measurable also at 18 months, and estimated at 24 months. The presence of a high number of IgA has also been detected, which indicates an adequate protection of the mucous membranes.

Vaccination notoriously provides immunogenic reaction against spike protein only and can only minimally elicit IgA increase. Therefore, the specific immune memory for SARS-CoV-2 in response to infection may theoretically be more comprehensive in recovered patients and it may persist in most subjects for up to one and a half to two years after infection. This data is promising for the prevention of both reinfection and especially of severe clinical pictures.

From the literature review it is therefore clear that, as already known from basic immunological notions, the cellular response is activated and remains even when the antibody response is no longer detectable. On the basis of these pathophysiology concepts, corroborated with a relevant number of emerging data from the most recent publications, some authors have proposed that the individuals recovered from a natural infection should be granted at least the same social status of COVID-19 immunity as people who have been fully vaccinated [102].

Regarding artificial immunity, it was repeatedly shown that it tends to decay more rapidly than the natural one and seems to be less effective in protecting from both infections and hospitalizations after 5–7 months. Conversely, regarding hybrid immunity, the increase in protection from infection conferred by vaccination of the recovered subjects is debated: the average value of infection reduction is valuable in terms of relative risk, as shown by the study of Shenai et al. [143] -RR = 1.82, whereas it is significantly lower as absolute risk-RA = 0.004 person-years. Furthermore, the greatest protection refers to an event (reinfection in those who have already overcome an infection) which is already uncommon in itself [184]; moreover, it is also debatable whether it is worthwhile to vaccinate a recovered person, considering that the clinical manifestations of a reinfection are milder than the first episode.

Fully vaccinated people who get COVID-19 seem to shed SARS-CoV-2 with viral loads similar to the unvaccinated individuals, thus literature data show little if no epidemiological benefit conferred by the vaccination in the recovered patients.

A large number of studies have clearly demonstrated that the available vaccines have not shown an adequate efficacy in protecting against infection (contagion). Furthermore, two-dose vaccinated people were shown to become more susceptible to infection than unvaccinated over the course of months. Lastly, preliminary data are showing that similar findings in vaccinated subjects also after three doses, especially over time, due to the decline in relative protection at longer follow-up.

Some of the studies included in this review have been carried out only in vitro, which may limit their value due to the lack of clinical findings. Moreover, some of them contain various biases, such as the lack or the insufficient assessment of a possible previous infection: consequently, the real rate of the subsequent reinfection is debatable in a few of the studies examined in this review. As regards reinfections, no clinical data emerge on the rates of asymptomatic and symptomatic cases, which is fundamental for identifying the real clinical need for vaccinating a recovered individual.

Finally, in most trials the investigated groups were not closed, hence patients could be transferred from one to another depending on the vaccination/infection status. Therefore, the accuracy of the follow-up findings and estimates may have been compromised.

Concerning safety of vaccination in the healed subjects, a re-appraisal of the current strategy is expected, since vaccine adverse reactions are regularly more intense in those who have overcome COVID-19 [225] in comparison to the side effects of vaccinated subjects without prior infection [229]. In these terms, the benefits/risks balance of hybrid immunity should also include the adverse reactions which take place in vaccinating healed subjects.

Concerning the incidence and the possible pathophysiology of the adverse events related to vaccination, a great number of studies have been published [231,232,233,234,235,236,237,238] although still, clear-cut evidence about their incidence and pathomechanisms does not exist.

Generally, in case of natural or vaccine-induced altered immunity it is expected an activation of several pro-inflammatory cascades, including assembly of inflammasome platforms, the response to type I interferon (IFN) and the subsequent nuclear translocation of NF-kB factor that follows, determining an up-regulation of these immunological pathways. Overall, these deregulated mechanisms are to be considered at the basis of several immune-mediated diseases which may occur after anti-COVID-19 vaccination, especially in genetically predisposed individuals.

In specifically predisposed subjects and/or due to specific issues related to vaccine content, it is expected that some ADE-based multi-organ (neuronal, myocardial, vascular cells primarily) detrimental interaction may occur in a quote of the recipients [222,239,240].

Newer strains are continuously emerging in this pandemic and the research about Omicron variants is still ongoing. It was shown that also the infection with Omicron variants confers host immunity. Due to the generally mild symptoms induced by Omicron variant and due to the natural immunity acquired after the infection, several authors have suggested that a kind of non-pharmacological mass “vaccination” is occurring through the latest virus variants, which may slow the spread of this complex pandemic [241]. Regarding Omicron variants, there are still some uncertainties. In fact, COVID-19 induced immunity mitigates the clinical manifestations of a re-infection with Omicron strain, hence the relative hospitalization rate is low. At the same time, vaccination shows a very low protective role against contagions with Omicron.

Regarding Omicron variants, there are still some uncertainties about the definition of best vaccination strategy. In fact, COVID-19 induced immunity mitigates the clinical manifestations of a re-infection with Omicron strain, hence the relative hospitalization rate is low. At the same time, vaccination shows a very low protective role against contagions with Omicron.

Our review aimed at evaluating the broad scenario of immunity in COVID-19 affected/vaccinated subjects. Beyond the circulating antibodies titre, which physiologically decrease over time (“hook effect”), we took into consideration also the adaptive memory immunity both of cellular and humoral type, that may potentially determine a more beneficial protection even on viral variants, as already demonstrated before the pandemic [242].

Of interest, also the complexity at the level of “random” antigenic recognition by APCs (Antigen presenting cells) on CD4 Helper cells should be carefully considered in these infected or vaccinated individuals. As with B cells, also T cells possess the cell receptor mechanism. Thus, an excess of antigen (such as in the case of repeated vaccinations) can “address” the gene rearrangement of these lymphocytes towards that specific antigen, losing their adaptability; this variation may result in an excessively specific adaptive immunity, hence losing its ability to combat for example ongoing viral variants (the so-called “self-organized critically theory” [221,243,244]. Basically, if the host’s immune system is overloaded with repeated exposure to the same antigen, reaching levels that exceed its stability limit, i.e., the self-organized criticality, there risk to develop autoimmunity increases.

In fact, it is expected that a new type of self-reactive CD4 helper T lymphocytes proliferates, which would give rise to autoimmune diseases more easily. Figure 4 (adapted from [244]) shows how the repetition of the exposure to the same antigen, rather than the usual exposure to various antigens, may determine the phenomenon described above.

Overall, our review has the limitation of having gathered a considerable amount of data derived from studies which may have inhomogeneous study-designs about different subtopics, which exposes to possible biases and flaws of the data analysis.

Anyway, the resulting findings seem to corroborate our conclusive speculations regarding the need for a revision of the vaccine strategy in the specifically examined population.

Overall, future studies are likely required to better define the still contradictory evidence regarding vaccination in the healed subjects, especially in case of Omicron variants. Objectively, the percentage of protection derived from a previous infection and/or from vaccination seems to vary considerably in terms of contagion, clinical manifestation, hospitalization, death rate. However, when reinfection occurs, the patients previously exposed to an antecedent variant of SARS-CoV-2 are protected from relevant clinical repercussions.

The potential role of vaccination against Omicron strain remains unclear as well, especially due to the typology of the currently employed vaccines, which more specifically targeting older variants of SARS-CoV-2.

## 6. Conclusions

In conclusion, the analysis of the literature regarding natural, post-COVID-19, immunity has highlighted a series of findings which indicate a good immunological protection in the vast majority of the individuals. The elicited natural immunity is typically of cell-mediated and humoral nature and it seems to protect against both reinfection and clinically serious illness.

Protective antibodies and memory B cells were found in many studies with follow-ups from 12 to 18 months after healing, and their presence was shown even more prolonged with the lengthening of observation times. Specifically, a piece of Swedish research with a follow-up after natural infection of up to 20 months showed a 95% protection rate from infection and 87% from hospitalization in those who have not added vaccinations [36].

From the biochemical and immunological point of view it has been clarified that the cellular response is activated and remains active also in absence of a detectable antibody response. More in detail, the presence of T CD4+ and CD8+ lymphocytes has been confirmed over time in subjects recovering from SARS-CoV-2 up to 18 months after infection. Vaccine-induced immunity proved to decay faster than natural (post-COVID-19) immunity, and the latter was the only type of immunological protection which is also activated by cross-reactivity towards other pathogens.

In general, it seems that previous SARS-CoV-2 infection provides greater protection than that offered by the single or double/triple dose vaccine. The risk of re-infection in post-COVID-19 subjects was documented as very low. For example, more than a year after the primary infection, unvaccinated people still have protection at around 70% (69% in a large cohort of UK health workers, [171]); of note, subsequent vaccination may raise this protection further.

In case of reinfection, the viral load has been calculated as about 10 times lower than that of a primary infection; similarly, the severity of the symptoms of reinfection is usually significantly lower than in the primary infection, with a lower degree of hospitalizations (0.06%) and almost no related deaths.

The protection from infection conferred by the vaccination cycle is very good after the first 14 days, however it tends to decline rapidly over the following months, nearly disappearing about five months after the second dose. Some literature data highlight that a later stage this vaccine-induced protection against contagion and/or serious illness becomes less evident than the one demonstrated in the unvaccinated individuals.

A few authors have reported that after being infected with SARS-CoV-2, the subjects are unlikely to benefit from COVID-19 vaccination [245,246]. Due to the documented prolonged immune response after COVID-19, the further administration of vaccine doses, especially from the second onwards, according to the majority of the studies does not lead to a significant improvement in immunity.

A series of possible dis-immunity-related pathomechanisms may occur due to the repeated vaccinations, as elicited above, hence the risk-to-benefit ratio in these cases seems to indicate no need for vaccine administration in this specific population of healed-from-COVID-19 subjects. Conversely a lower level of evidence is available for the literature about the efficacy of hybrid immunity, as the results of the studies are sometimes contradictory.

Further investigation is needed to evaluate statistically significant benefits conferred by hybrid immunity, considering that post-vaccine local and systemic adverse events are 40% and 60% higher, respectively, in exposed subjects with a previous history of SARS-CoV-2 infection [227]. When taking into consideration future vaccination strategy, WHO indicted the need for a rapid update of the currently available vaccines; similarly, in the after-COVID-19 subjects, in view of the mild clinical manifestations of the reinfections and of Omicron infections, updated therapeutic and epidemiological strategies could be developed.

Overall, in view of the data presented above through the present narrative review, vaccination of the recovered individuals should be re-evaluated, since they seem to show a more effective and lasting natural immunity compared to the vaccine-induced one, as is already known for other infectious diseases. In conclusion, our review has examined a large group of scientific studies which mostly demonstrate the value of the natural COVID-19-induced immunity, favorably comparing to the vaccine-induced immunological protection under several points of view.

Future research on these topics may desirably further elucidate a few critical points such as: (a) quantification of the durability of natural immunity over time; (b) evaluation the impacts of Omicron latest variants on both types of immunity, (c) assessment of the hybrid immunity with reference to short/mid-term protection, (d) stratification of the risk/benefit ratio in the possible candidates to vaccine.

Most likely, besides the evidence exposed in this extensive narrative review, the establishment of the individual’s immunological (cellular and humoral) profile towards the SARS_CoV-2 would help tailor a better decisional preventive/therapeutic process, always in combination with the clinical picture and anamnestic background of the patient.

## Figures and Tables

**Figure 1 jcm-11-06272-f001:**
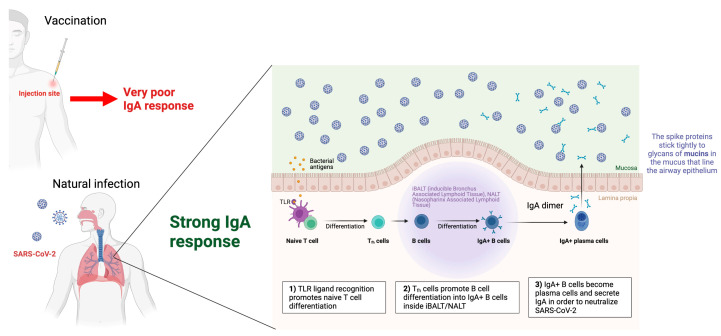
Natural infection leads to greater production of lgA compared to vaccination. IgA is produced in the mucous membranes through the activation and maturation of B lymphocytes and consequently of plasma cells in subjects naturally infected by SARS-CoV-2. On the other hand, vaccination only minimally elicits the IgA production in the mucous membranes.

**Figure 2 jcm-11-06272-f002:**
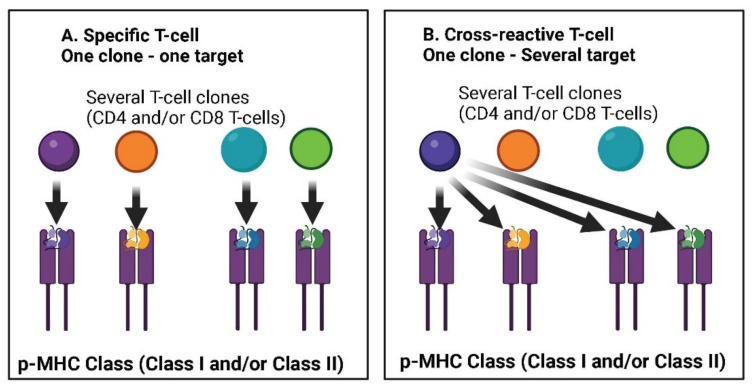
T-cell cross reactivity (Heterologous adaptive T cell-mediated immunity). (**A**). Each colored sphere represents a specific T cell which is directed to a corresponding antigen presented on the MHC. (**B**). One or more T cell can develop (or it could already exist) a cross-reactivity to other antigens.

**Figure 3 jcm-11-06272-f003:**
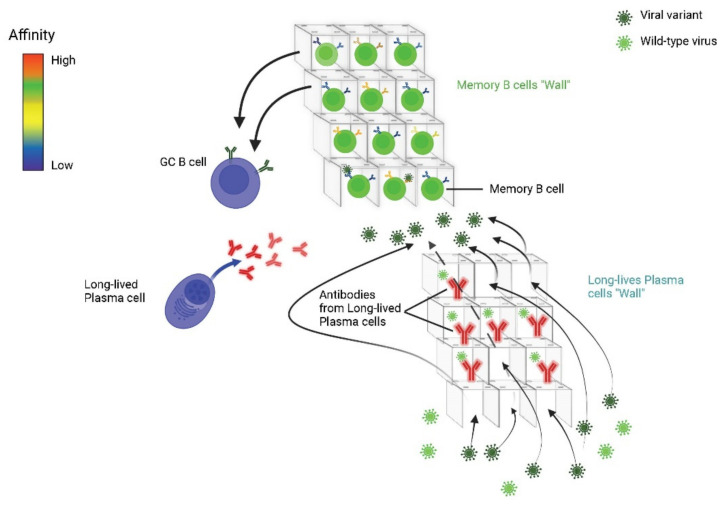
Humoral immunity naturally produces memory B cells and antibodies that also recognize viral variants. The long-lived plasma cells in the bone marrow secrete highly selected antibodies, specific and related to the first form selected. They are specific to the encountered pathogen (represented in red) and are against the relative infection. Variant pathogens (e.g., other viruses) can find “holes” in this barrier; however they may encounter a second barrier made up by memory B cells that were less selected. This happens because memory B cells retain a wider range of antigenic affinity and adaptability. Memory B lymphocytes are activated by the different variants; in this case, they either differentiate into long-lived plasma cells or to return to the germinal centers (GCs) in order to replenich the pool of memory B cells.

**Figure 4 jcm-11-06272-f004:**
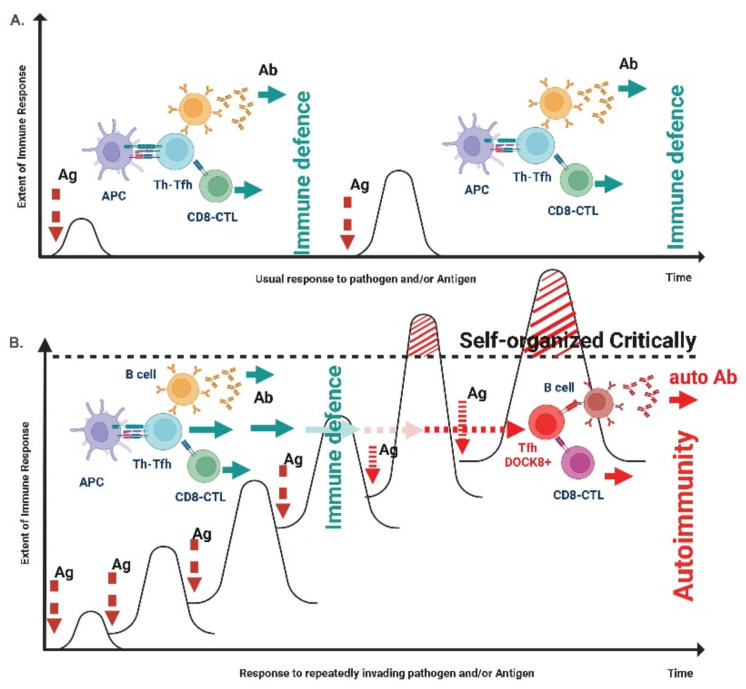
Self-organized Critically Theory. Each system remains in equilibrium when each trigger is modulated, maintaining homeostasis. In image (**A**). we see how the humoral and cellular adaptive immune response is perfectly balanced by a not excessive stimulation by an antigen. The result is an optimal and balanced (regulated) response. In image (**B**). the system undergoes multiple stresses for which it is unable to balance itself, effectively creating an isotype of follicular helper T lymphocytes (DOCK8+, Dedicator of Cytokinesis 8). These lymphocytes give instructions both to the humoral and to the cytotoxic effector compartment, determining the fate of the immune response. In this scenario, the answer is not balanced, but it is rather distorted and leads to an inversion of the adaptive activity, predisposing these lymphocytes to the search of antigens previously encountered, therefore to autoimmunity.

**Table 1 jcm-11-06272-t001:** Summary of included studies and primary outcomes results about the risk of reinfection.

Study	Population	Follow-Up	Outcomes
Abu-Raddad et al., 2021 [130]	General population N = 43.044 antibody-positive at baseline	Median: 114 days Maximum: 242 days	Risk of reinfection: 0.1% (95% CI 0.08%–0.11%)
Crawford NW, 2022 [131]	General population N = 688.418 PCR positive at baseline	515 days	Risk of reinfection by age: <5 years 0.18% 5–11 years 0.24% 12–16 years 0.49% >16 years 0.73%
Dos Santos et al., 2021 [122]	Healthcare workers N = 378 qRT-PCR positive at baseline	Median: 41 days Maximum: 130 days	Risk of COVID-19 recurrence: 7.9% (both reappearance of the same virus and new infections) 1 Virus genome sequencing identified reinfection (0.26%)
Flacco et al., 2021 [119]	General population N = 7173 PCR positive at baseline	Median: 201 days Maximum: 414 days	Risk of reinfection: 0.33% Risk of hospitalization: 0.06% Risk of lethal events: 0.01%
Hall et al., 2021 [117]	Healthcare workers N = 6.614 antibody-positive at baseline	Median: 202 days Maximum: 227 days	Adjusted odds ratio of probable reinfection: 0.1 (95% CI 0.00–0.03)
Hanrath et al., 2020 [12]	Healthcare workers N = 1.038 PCR or antibody-positive at baseline	Median: 173 days Maximum: 229 days	Symptomatic reinfection: 0% (95% CI 0%-0.4%)
Hansen et al., 2021 [132]	General population N = 11.068 PCR positive at baseline	Median: 122 days Maximum: 295 days	Relative risk: 0.20 (0.16–0.25)
Harvey et al., 2020 [9]	General population N = 378.606 PCR positive at baseline	>90 days after first infection	Risk of reinfection: 0.3% Relative risk: 0.10 (95% CI 0.05–0.19) declining over time
Houlihan et al., 2020 [11]	Healthcare workers N = 33 antibody-positive at baseline	90 days	1 PCR positive on days 8 and 13 after enrolment (probable reappearance of the same virus)
Jeffery-Smith et al., 2021 [114]	Staff & residents at care homes N = 88 PCR or antibody-positive at baseline	120 days	Risk of reinfection: 1.1% Relative risk: 0.04 (95% CI 0.005–0.27)
Krutikov et al., 2021 [121]	Staff & residents at care homes N = 634 antibody-positive at baseline	Median: 79 days Maximum: 300 days	Relative adjusted hazard ratios (any reinfection): Residents of care home: 0.15 (0.05–0.44); Staff of care home: 0.39 (0.19–0.82)
Leidi et al., 2022 [120]	General population N = 498 antibody-positive at baseline	Median: 35,6 weeks Maximum: 38,8 weeks	Risk of reinfection: 1%
Letizia et al., 2021 [123]	Marines N = 189	6 weeks	Risk of reinfection: 10% Relative risk: 0.45 (95% CI 0.32–0.65)
Lumley et al., 2021 [116]	Healthcare workers N = 1.265 antibody-positive at baseline	Median: 139 days Maximum: 217 days	Risk of reinfection: 0.16% Relative risk: 0.11 (95% CI 0.03–0.44)
Mishra et al., 2021 [128]	General population N = 1170 antibody-positive at baseline	Median: 258 days Maximum: 319 days	Risk of reinfection: 0.26% Relative risk: 0.023 (95% CI: 0.007–0.073) Risk of hospitalization: 0.08% Risk of lethal events: 0%
Perez et al., 2021 [133]	General population N = 149.735 PCR positive at baseline	Median: 165 days Maximum: 325 days ca.	Risk of reinfection: 0.1%
Pilz et al., 2021 [134]	General population N = 14.840 PCR positive at baseline	Median: 210 days Maximum: 300 days	Risk of reinfection: 0.27% Relative risk: 0.09 (95% CI: 0.07–0.13)
Qureshi et al., 2022 [135]	General population N = 9119 positive	Median: 116 days Maximum: 137 days	Risk of reinfection: 0.7% (95%, CI: 0.5%-0.9%) declining over time
Sheehan et al., 2021 [136]	General population N = 8.845 PCR positive at baseline	90 days after first infection	Protective effectiveness (any reinfection): 78.5% (95% CI: 72.0%–83.5% growing over time
Vitale et al., 2021 [118]	General population N = 1597 PCR positive at baseline	Median: 280 days Maximum: 321 days	Risk of reinfection: 0.31%; (95% CI, 0.03%-0.58%) Risk of hospitalization: 0.06% Risk of lethal events: 0%

## Data Availability

Not applicable.

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
