# Peer review of "SARS-CoV-2—The Role of Natural Immunity: A Narrative Review"

_jcm, 2022, doi:10.3390/jcm11216272_

Round 1

Reviewer 1 Report

This manuscript reviews the literature about the duration of natural and post-vaccination immunity to SARS-CoV-2 reinfection and incidence of adverse events in recovered patients. Overall, the paper covers a number of meta-analyses and other work on reinfection. Aside from some minor editing.

Minor Comments:

Abstract: "Omicron-5" in abstract should probably be corrected to either BA.5 or Omicron.

Materials and Methods: in vitro and in vivo should be in italics.

Author Response

Response to Reviewer 1 Comments

"This manuscript reviews the literature about the duration of natural and post-vaccination immunity to SARS-CoV-2 reinfection and incidence of adverse events in recovered patients. Overall, the paper covers a number of meta-analyses and other work on reinfection. Aside from some minor editing."

  1. Abstract: "Omicron-5" in abstract should probably be corrected to either BA.5 or Omicron: In the abstract we modified Omicron-5 with Omicron BA.5
  2. Materials and Methods: in vitro and in vivo should be in italics. We put the words in italics.
  3. Moderate English changes required. 

    We ameliorated the English language in our paper. We let an English native speaker checking and correcting the document.

Thank you for your supportive review. 

Reviewer 2 Report

After reviewing the document, I consider it to be an excellent review on one's own immunity to covid infection and that provided by vaccination, in addition to criticizing some important points and proposing ideas about vaccination.

Author Response

Response to Reviewer 2 Comments

"After reviewing the document, I consider it to be an excellent review on one's own immunity to covid infection and that provided by vaccination, in addition to criticizing some important points and proposing ideas about vaccination."

English language and style are fine/minor spell check required.

We ameliorated the English language in our paper. We let an English native speaker checking and correcting the document.

Thank you for your enthusiastic and very encouraging review.

Reviewer 3 Report

1.       There should be a section, on the future direction which is important for this article.

2.       The authors should also describe the limitations in the present scenario regarding COVID-19.

3.       The introduction is needed a serious extension focusing on the current issue of the manuscript.

4.       Authors should describe the host factors and natural immunity boosters such as probiotics regarding the theme of this manuscript. Please follow the mentioned references.

https://pubmed.ncbi.nlm.nih.gov/33425362/

https://www.frontiersin.org/articles/10.3389/fcimb.2022.928704/full

5.       This is a review article. The material and methods section is not appropriate. Please remove.

6.       Authors need to discuss secondary infections also, about the natural immunity. There are many good reports available. Please check and update.

7.       This is a review article and the reviewer recommends including interactive illustrations relevant to the current theme of the manuscript. 

Author Response

Point 1: There should be a section, on the future direction which is important for this article.

Response 1: we have integrated the future perspectives of this research into the conclusions.

Point 2: The authors should also describe the limitations in the present scenario regarding COVID-19.

Response 2: we inserted the limits of our narrative review in the discussion, in particular about all the different study designs that we analyzed throughout the literature and the consequent limitations.

Point 3: The introduction is needed a serious extension focusing on the current issue of the manuscript.

Response 3: we strongly extended the introduction, inserting both contextualization concerning the immune system and the main topics of our review.

Point 4: Authors should describe the host factors and natural immunity boosters such as probiotics regarding the theme of this manuscript. Please follow the mentioned references.

Response 4: We wrote about immunity boosters such as probiotics and entered the suggested references in the section 4.3.

Point 5: This is a review article. The material and methods section is not appropriate. Please remove.

Response 5: Regarding the material and methods section, although the reviewer asked to remove it, we think it is a fundamental section to expose the sources and methods by which we investigated the pertinent literature. Furthermore, we put this section in the paper as required by the editor in the Journal instructions (Available at https://www.mdpi.com/journal/jcm/instructions). 

Point 6: Authors need to discuss secondary infections also, about the natural immunity. There are many good reports available. Please check and update.

Response 6: We wrote about the secondary infections in the section 4.3.

Point 7: This is a review article and the reviewer recommends including interactive illustrations relevant to the current theme of the manuscript.

Response 7: We have created four figures relevant to the current themes of the article.

Extensive editing of English language and style required.

We ameliorated the English language in our paper. We let an English native speaker checking and correcting the document.

Thanks for the supportive and ameliorative review.

Round 2

Reviewer 3 Report

The authors updated the manuscript as per the reviewers' comments and suggestions.